# Integration of 3D Hydrodynamic Focused Microreactor with Microfluidic Chemiluminescence Sensing for Online Synthesis and Catalytical Characterization of Gold Nanoparticles

**DOI:** 10.3390/s21072290

**Published:** 2021-03-25

**Authors:** Yanwei Wang, Michael Seidel

**Affiliations:** Institute of Hydrochemistry, Chair of Analytical Chemistry and Water Chemistry, Technical University of Munich, Elisabeth-Winterhalter-Weg 6, 81377 Munich, Germany; yanwei.wang@tum.de

**Keywords:** gold nanoparticles, online, chemiluminescence, catalyst characterization, microfluidic chip

## Abstract

Chemiluminescence assays have shown great advantages compared with other optical techniques. Gold nanoparticles have drawn much attention in chemiluminescence analysis systems as an enzyme-free catalyst. The catalytic activity of gold nanoparticles for chemiluminescence sensing depends on size, shape and the surface charge property, which is hard to characterize in batches. As there is no positive or negative correlation between chemiluminescence signals and sizes of gold nanoparticles, the best way to get optimal gold nanoparticles is to control the reaction conditions via online chemiluminescence sensing systems. Therefore, a new method was developed for online synthesis of gold nanoparticles with a three-dimension hydrodynamic focusing microreactor, directly coupled with a microfluidic chemiluminescence sensing chip, which was coupled to a charge-coupled device camera for direct catalytical characterization of gold nanoparticles. All operations were performed in an automatic way with a program controlled by Matlab. Gold nanoparticles were synthesized through a single-phase reaction using glucose as a reducing agent and stabilizer at room temperature. The property of gold nanoparticles was easily controlled with the three-dimension microreactor during synthesis. The catalyst property of synthesized gold nanoparticles was characterized in a luminol–NaOCl chemiluminescence system. After optimizing parameters of synthesis, the chemiluminescence signal was enhanced to a factor of 171. The gold nanoparticles synthesized under optimal conditions for the luminol–NaOCl system were stable for at least one month. To further investigate the catalytic activity of synthesized gold nanoparticles in various situations, two methods were used to change the property of gold nanoparticles. After adding a certain amount of salt (NaCl), gold nanoparticles aggregated with a changed surface charge property and the catalytic activity was greatly enhanced. Glutathione was used as an example of molecules with thiol groups which interact with gold nanoparticles and reduce the catalytic activity. The chemiluminescence intensity was reduced by 98.9%. Therefore, we could show that using a microreactor for gold nanoparticles synthesis and direct coupling with microfluidic chemiluminescence sensing offers a promising monitoring method to find the best synthesis condition of gold nanoparticles for catalytic activity.

## 1. Introduction

Chemiluminescence (CL) is a phenomenon in which a specific molecule gets energy from a redox reaction and is excited. The molecule emits light when it returns to a ground state. As there is no need of an excitation source and optical filters, it shows great advantages compared with other optical techniques, such as low cost, simple instrumentation and easy automation [1]. Moreover, the high sensitivity, and wide linear range make CL-based assays applicable in different scientific and industrial areas [2]. Enzymes are used in most CL methods as a catalyst, such as horseradish peroxidase and alkaline phosphatase [3]. However, enzymes have the disadvantages of short lifetime and low stability, which means they are easily denaturalized. Moreover, enzymes are usually expensive with a complex labeling procedure [4]. Recently, precious metal nanomaterials have participated in the CL reaction, which improved performance of the CL system [5]. Among these nanomaterials, gold nanoparticles (AuNPs) have attractive properties, such as easy preparation and modification, stability, large surface area to volume ratio, and good biocompatibility, which have attracted researchers’ attention [6]. Some researchers have applied AuNPs as a catalyst in a CL system [7,8,9]. The catalytic activity of AuNPs with different sizes was systematically investigated in luminol–H_2_O_2_ CL reactions by Cui’s group [10]. It was found that AuNPs with different sizes had different enhancement of the CL signals. However, there is no positive or negative correlation between CL signals and sizes of AuNPs. In other CL systems, studies have also shown that there was no direct relationship between the size of AuNPs and signal intensity [7,8,10,11,12,13,14]. Additionally, the optimal size of AuNPs depends on different CL systems [7,8,10,11,12,13,14]. For example, in the luminol–H_2_O_2_ system, the 38 nm AuNPs showed best catalytic performance among the tested AuNPs [10]. However, in luminol–ferricyanide and luminol–hydrazine CL systems, the most intensive CL signal was obtained by using 25 and 15 nm AuNPs, respectively [7,11]. Due to the limited number of AuNPs tested and various CL systems, it is impossible to determine the best AuNPs with a specific size. In addition, the morphology of AuNPs affects their catalytic performance in CL reactions. Aggregated AuNPs were found to induce a higher signal in CL reactions with luminol than dispersed ones [15,16,17]. Besides their size and shape, AuNPs with different surface charge properties also display various functions in CL reactions [16]. Cationic AuNPs or AuNPs with lower negative charge density have been proven to exhibit higher catalytic activity [16,17,18]. Due to all these variables, it is difficult to confirm the optimal AuNPs for a certain CL reaction. Although the most effective AuNPs for a specific CL reaction have not been determined, AuNPs have been successfully used in many practical applications, as shown in Table 1. Even though 38 nm AuNPs have been proven to show better catalytic activity in the luminol–H_2_O_2_ CL system [10], they were not used in any of these applications. According to our knowledge, researchers only buy commercial AuNPs or produce them batch-wise. On the one hand, no one can determine which AuNPs exhibit the best catalytic activity. On the other hand, there is no simple way to control the size of AuNPs during the synthesis process. Therefore, it is crucial to develop an online monitoring system which can easily control the property of AuNPs during synthesis and immediately inspect the catalytic CL activity.

Microreactors have been exploited by many research groups to achieve rapid and tunable synthesis of nanoparticles [23,24,25,26], fulfilling the promise of automated systems for reaction optimization [27]. Microfluidic reactors were shown to synthesize products with narrower size distributions and faster reaction rates compared to conventional batch synthesis [23]. Extremely short mixing times can be achieved in microfluidic reactors due to the thin fluid layer thickness [28]. Moreover, decreased amounts of reagents are used and limited by-products are generated. Microreactors can even be applied for automated multi-step synthesis [29]. A continuous flow microreactor was used for the synthesis of AuNPs by Wagner et al. [23]. Significant fouling of the micro-channels was observed due to deposition of nanoparticles on the reactor walls. Fouling can be relieved by using focusing flow in which sheath streams can insulate nanoparticles from channel walls. Moreover, focusing the reaction flow allows formation of more monodispersed particles due to the uniformity of concentration, residence time and fluid velocity in the center [30]. Recently, our group invented a flow-focused microreactor for synthesis of magnetic nanoparticles [31]. The laminated three-dimension (3D) flow-focused microreactor was further developed for synthesis of AuNPs without fouling [32]. This continuous synthesis method has the potential to couple online with analytical instruments like inductively coupled plasma mass spectrometry (ICP-MS), flow-based UV–Vis spectrometers or sensors. For CL sensing, continuous flow injection improves mixing between the luminol and oxidant, which can result in a higher intensity of CL signal than in a cuvette [9]. Flow-based methods allow a continuous light emission if luminol; oxidants and catalyst are pumped into the microfluidic chip constantly. The CL signal in microchannels can be imaged by a CCD camera for sensing applications.

The principle of online synthesis using a 3D hydrodynamic focused microreactor combined with microfluidic CL sensing was shown for the first time in this work. The catalytical characterization of AuNPs produced continuously through a single-phase reaction were directly measured by CL reactions of luminol and NaOCl in microfluid channels by a CCD camera. NaOCl was one of the first reagents used to demonstrate luminol CL with brilliant blue emission, and is a convenient choice in the classroom [33]. The method can be applied to other CL systems in the same way. AuNPs were synthesized by the reduction of tetrachloroaurate (III) ions with glucose as reducing agent and stabilizer and sodium hydroxide (NaOH) adjusting pH [34]. Synthesizing AuNPs with glucose has never been tried in a 3D microreactor before. Glucose has the advantage that AuNPs can be synthesized at room temperature, unlike with citrate solutions where heating is required [35]. Moreover, compared with the strong reducing agent sodium borohydride (NaBH_4_), glucose is non-toxic and cannot react with water. The pH value and concentration of glucose has an important effect on the size distribution and stability of the nanoparticles [36]. Therefore, the sizes and properties of AuNPs were modulated by varying the concentration of reagents to obtain optimal catalytic activity in enhanced CL detection by luminol and NaOCl. The stability of AuNPs synthesized under optimal conditions was checked for one month. The effect of aggregation and the surface property of AuNPs on catalytic activity were also monitored by CL imaging after adding salt and molecules with thiol groups. A possible mechanism for luminol–NaOCl–AuNPs is shown in Scheme 1. AuNPs show catalytic activity with facilitating radical generation and electron transfer processes on the surface of the AuNPs [10]. AuNPs were stable with negative charged repulsion. When salt was added, the repulsion was screened with decreased density of negative charge which resulted in the aggregation of AuNPs. Therefore, anions can easily interact with the surface of AuNPs and the aggregated AuNPs show better catalytic performance than the dispersed ones [17]. Some organic compounds with -SH groups could strongly interact with AuNPs, and the amount of radical absorbed on the surface of AuNPs might be reduced, which might result in the inhibitive phenomenon [37].

## 2. Materials and Methods

### 2.1. Materials

Gold (III) chloride trihydrate (HAuCl_4_ · 3H_2_O, ≥99.9% trace metal basis, Sigma-Aldrich, Munich, Germany) was used as a gold precursor and D-glucose (Sigma-Aldrich, Munich, Germany) was used as a reductant with the aid of sodium hydroxide (NaOH, reagent grade, ≥98%, pellets, Sigma-Aldrich, Munich, Germany). Luminol stock solution (4 × 10^−2^ M) was prepared by dissolving 3-aminophthalhydrazide (luminol, Sigma-Aldrich, Munich, Germany) in 0.10 M NaOH and stored in a fridge for one week before use. The stock solution was diluted with ultrapure water to get the specific concentration. Sodium hypochlorite (NaOCl, 12% Cl) supplied by Carl Roth (Karlsruhe, Germany) was diluted for the working solution. Hydrochloric acid (HCl, Sigma-Aldrich, Munich, Germany, ACS reagent) was used to clean the microreactor after synthesis. Stock solution of glutathione (0.01 M) was prepared by dissolving glutathione (Sigma-Aldrich, Munich, Germany) in ultrapure water, and it was diluted by ultrapure water to make a working solution. Ultrapure water was used to prepare all aqueous solutions. The poly (methyl methacrylate) (PMMA) sheets with a thickness of 0.2 mm were supplied by the company Modulor Material Total (Berlin, Germany). The double-sided pressure-sensitive adhesive (PSA) tape (ARcare 90106) was supplied by Adhesive Research (Glen Rock, PA, USA). The carrier sheet with a thickness of 10 mm was fabricated by our in-house workshop. The fabricated PMMA carrier contained threads (1/4”—28 UNF) to allow the connection with PTFE tubing.

### 2.2. Synthesis of AuNPs in 3D Microreactor

A laminated 3D hydrodynamic flow-focused microreactor was constructed with PSA tape and PMMA. The method eliminated cleanroom requirements due to the absence of lithographic process. The difficulty in designing and fabricating a 3D microreactor was solved by the simple assembly process of layering precut sheets. This 3D microreactor has been proven to synthesize reproducible AuNPs without fouling [32]. The microreactor applied for synthesis of AuNPs was constructed of seven layers of PMMA sheet and PSA tape with 3 inlets for HAuCl_4_, glucose and NaOH, and one outlet for AuNPs as shown in Figure 1. A series of different concentrations of NaOH (10^−3^–1 M) and glucose (3 × 10^−4^–3 × 10^−1^ M) was tested to see at which concentration AuNPs were generated.

### 2.3. Procedures for CL Measurements

The CL measurements were conducted on CCD camera (16-bit) with a microchip insert for direct online CL imaging as depicted in Figure 2. The microchip was composed of a transparent glass sheet for coving, one PSA layer and one black plastic sheet with holes for inlets and outlet. The PSA layer with channel offers a place for reagents mixing and CL generation. The glass allows generated light emissions to pass through, and the light is recorded by a sensitive CCD camera with an image. CL intensity is described as the gray intensity of each pixel, ranging from 0 to 65,536 au. The background from the dark signal of the CCD was recorded before the measurement, and the background was subtracted from each image before evaluation. ImageJ was used to integrate the signal intensity over the pixels.

### 2.4. Automated Synthesis and Online CL

The luminol–NaOCl system was coupled with online synthesized AuNPs for catalyst characterization. All operations, including synthesis of AuNPs and CL generation in the coupled CCD camera, were performed in an automatic way. All components were connected using PTFE tube (inner diameter 0.8 mm). The combined setup is shown in Figure 3. Reagents were supplied to the microreactor with three glass syringes (Innovative Laboratory Systems GmbH, Stutzerbach, Germany) connected to a 6-port valve (Cavro Smart Valve, Tecan Group Ltd., Männedorf, Switzerland). The first ports of each valve were used for intake of glucose, NaOH and HAuCl_4,_ which were reagents for synthesis of AuNPs. Then they were transferred to the 5th ports, which were connected to the inlet of the microreactor. The second ports were for injection of water to clean syringes. The 6th ports were for outlet of waste during or after synthesis. The 3rd port of the first valve was for cleaning solution after synthesis. The glass syringes were operated by three custom-made pumps (GWK Precision Technology, Munich, Germany) which were controlled with Matlab (The MathWorks, Inc., Natick, MA 01760, USA) by a host computer connected via an Ethernet cable. The synthesized gold nanoparticles could be directly pumped into a microfluidic CL sensing chip. Another pump was used to deliver reagents for changing the property of AuNPs, luminol and NaOCl, and the flow rates were the same at 1 µL/s. In this paper, NaCl and glutathione were used as two kinds of property-changing reagents. When they were not applied, the syringes for them were blocked. The property-changing reagents were mixed with AuNPs through a three-way valve. The solution from the three-way valve was then mixed with luminol by another three-way valve. NaOCl was then mixed with the mixture in the microchip inserted in the CCD camera to generate a CL signal. With all reagents pumped into the microchip simultaneously, the signal could be recorded with the CCD camera online.

### 2.5. Offline Characterization of AuNPs

UV–Vis absorbance spectra for gold nanoparticle suspensions were recorded using a SPECORD 250 PLUS UV–Vis spectrometer (Analytik Jena, Jena, Germany). The spectrometer uses deuterium and halogen lamps to produce UV light and a visible range of electromagnetic wavelengths. A beam of light with a wavelength ranging from 400 to 900 nm was used for measurement. Disposable polystyrene cuvettes (UV cuvette semi-micro, 1.5–3.0 mL, Brand GmbH, Hamburg, Germany) were used for containing samples and reference. Ultrapure water was used as a reference sample as all solutions were prepared in ultrapure water. Scanning electron microscopy (SEM) images were acquired on a model Sigma 300 VP microscope (Zeiss Gemini, Graz, Austria). The samples were analyzed with an InLens detector using an acceleration voltage of 10 kV, a 30 µm aperture and a working distance of about 1.4 mm.

## 3. Results and Discussion

### 3.1. Effects of the Reagent Concentrations on AuNPs Synthesis

The concentration of reagents is an important factor in synthesis of AuNPs. Neither high nor low concentration of NaOH was suitable for synthesis of AuNPs, as shown in Figure 4a. When the pH was low with 10^−3^ M NaOH, there were no surface plasmon resonance (SPR) absorption bands from 400 to 900 nm, which indicated that no AuNPs were generated [38]. Since OH^−^ participates in the reduction reaction, the pH environment has a significant impact on the synthesis of AuNPs, and glucose can only act as an effective reducing agent in the presence of OH^−^ [34]. With high concentration (10^−1^ M and 1 M), there were inconspicuous broad peaks. This implied that AuNPs were aggregated. AuNPs were stabilized with glucose via electrostatic repulsions. High concentration of ions from NaOH may break the electrostatic balance, which can cause aggregation of AuNPs [39]. When the concentration of NaOH rose to 10^−2^ M, the absorbance peak with small full-width at half maximum (FWHM) of the SPR band and high intensity appeared, which indicated narrower size distribution and higher concentration of AuNPs. Therefore, the concentration of NaOH should be around 10^−2^ M. The effect of glucose concentration on the UV–Vis absorption spectra of AuNPs synthesized with 10^−2^ M NaOH is shown in Figure 4b. For very low concentration of glucose (3 × 10^−4^ M), limited AuNPs were produced as there was only a small peak around 550 nm. In this case, the amount of glucose was not enough to reduce all Au^3+^ ions in the solution. With higher glucose concentrations from 3 × 10^−3^ M to 3 × 10^−1^ M, more AuNPs were produced. The surface-plasmon maximum absorption wavelength (λ_max_) of AuNPs shifted to a lower value (from 578 to 529 nm) with decreasing size of AuNPs. Therefore, it is easy to infer that the higher the glucose concentration, the more effective the reduction function of glucose. There would be more nucleation sites which result in smaller AuNPs and higher number density. To make sure all Au ions can be reduced, concentration of glucose should be higher than 10^−2^ M. The concentration of glucose can be tuned to obtain various sizes of AuNPs.

### 3.2. Effect of Synthesis Parameters on Catalytic Property of AuNPs

The concentration of glucose and NaOH can affect the size distribution of AuNPs [36]. AuNPs with different sizes can further affect the CL signals with different catalytic activity [10]. As there is no positive or negative correlation between CL signals and sizes of AuNPs, the best way to get optimal AuNPs is to control the reaction conditions via online CL sensing. The properties of AuNPs were easily controlled with the 3D microreactor during synthesis. The parameters for synthesis were optimized for the luminol–NaOCl–AuNPs system to obtain the highest CL signal. To test the catalytic activity of AuNPs, the tube for property-changing reagents was removed and AuNPs were mixed directly with luminol. The effect of NaOH concentration was tested in the range of 0.5 to 10 mM with 0.01% HAuCl_4_ and 0.3 M glucose (Figure 5a). As shown in the figure, the intensity of CL increased with the increase of NaOH concentration in the range of 0.5–2 mM. The CL intensity decreased after concentration of NaOH reached 2 mM. After that, the CL signal reach another maximum and then decreased again. There was no linear relationship between NaOH concentration and CL signal; 2 mM NaOH was chosen for further application as it can offer the highest CL signal at these conditions. The effect of glucose concentration on intensity of CL was studied, ranging from 0.2–1 M (Figure 5b). There was a steady increase of CL intensity as concentration of glucose increased. As glucose solution is almost saturated when the concentration is 1 M, 1 M glucose was used for further application. The effect of HAuCl_4_ concentration was also investigated as shown in Figure 5c. When the concentration of HAuCl_4_ solution was lower than 0.05%, the intensity of CL increased with the increase of HAuCl_4_ concentration. The reason could be that more AuNPs were generated with higher concentration of HAuCl_4_ solution. When the concentration of HAuCl_4_ solution was higher than 0.05%, the CL intensity decreased because the properties of AuNPs changed. Considering the CL intensity, the ideal conditions for synthesis were as follows: 2 mM NaOH, 1M glucose and 0.05% HAuCl_4._ Under the optimal conditions, the synthesized AuNPs were quasi-spherical with a diameter of 15.32 ± 1.09 nm, as shown in Appendix A.

After getting the optimized conditions for synthesis of AuNPs, it was still not clear if the synthesized AuNPs or the reagents for synthesis acted as catalysts. The comparison of CL signals from background and all blanks were done as shown in Figure 6. The background was from the totally dark signal without luminol–NaOCl. Since the background signal was subtracted from each image before the evaluation, the signal for background was 0 (Figure 6a). When luminol and NaOCl were mixed in the chip and only water was added instead of AuNPs, a gray line was shown in the image with a signal of 235 ± 2, as shown in Figure 6b. The most intensive CL signal was obtained for synthesized AuNPs with a signal of 40,378 ± 367 and there was a bright line as shown in Figure 6c. The enhancement was 171 times. Blank experiments were also carried out, including with NaOH, glucose and HAuCl_4_ solutions with the same concentrations for synthesis. As shown in Figure 6, there was no significant enhancement of CL signal when HAuCl_4_ was used and the signal reach 245 ± 10 as the concentration was too low, whereas NaOH and glucose could enhance the luminol CL signal slightly to 575 ± 7 as pH was changed. This effect is described elsewhere, too [10]. The concentration of unreacted NaOH and glucose in gold colloids was too low to make a contribution to the enhancement of CL intensity. Thus, the catalytic activity of gold colloid was attributed to AuNPs rather than due to NaOH, glucose or HAuCl_4._

### 3.3. Stability of Synthesized AuNPs

The stability of AuNPs can be estimated by observing the color of colloid solution, which did not change with time. The stability of the synthesized AuNPs was carefully investigated over one month by UV−Vis spectroscopy. Figure 7 shows the UV–Vis spectra of AuNPs on day 1 (right after the synthesis), day 3, day 6 and day 31. There was only a slight shift of the SPR band (from 552 to 554 nm) from day 1 to day 3 with a decreased absorbance from 0.95 to 0.93. After that, the SPR band did not change until day 31. The size of AuNPs was maintained and aggregation did not take place.

### 3.4. Effect of Property-Changing Reagents on Catalytic Property of AuNPs

Although the synthesized AuNPs were stable, aggregation could be caused by adding salt. The aggregation of AuNPs can change the CL intensity in a luminol system and many analytical methods were developed according to this principle [17,20,40,41]. Here, different concentrations of NaCl were added to AuNPs to change the property and CL signals were recorded to monitor the catalytic activity as shown in Figure 8a. With increased concentration of NaCl, the CL signal was enhanced significantly. UV–Vis absorption spectra analysis was carried out to inspect the property change after adding salt. As shown in Figure 8b, without salt addition, the absorption was high and SPR band of AuNPs was narrow, indicating the AuNPs were highly dispersed. As the salt concentration increased, the intensity of SPR decreased with a lower number concentration of AuNPs. Moreover, the absorption spectra were broader and shifted to a high value, indicating a bigger size of AuNPs caused by aggregation. With enough salt, the negative charged repulsion between AuNPs was screened and AuNPs aggregated [17]. Luminol–NaOCl CL reaction occurred in alkaline solution and the main molecules in the reaction were hypochlorite ion (OCl^−^) and luminol anion. The anionic molecules would not easily interact with the AuNPs because of a negative charge on the surface. The aggregated AuNPs with lower negative charge density caused by adding salt would be favorable for adsorption and electron transfer. Therefore, the catalytic activity of aggregated AuNPs caused by salt is better than that of dispersed AuNPs.

There are some organic compounds containing -OH, -NH_2_ or -SH groups which were reported to easily interact with AuNPs and change the surface property [42]. Glutathione was used as one example of a property-changing reagent with -SH groups. The CL signal was significantly inhibited with glutathione, indicating the decreased catalytic activity of AuNPs, and the degree of inhibition was related to the concentration of glutathione, as shown in Figure 9. With a concentration of 1 mM, glutathione can inhibit 98.9% of the CL signal, from 19,977 to 211. In the luminol–NaOCl–AuNPs system, there are some oxygen intermediates. A decrease of CL intensity can be caused by the competition between reducing groups of -SH and luminol for active intermediate radicals [17]. The surface of AuNPs was occupied by the compounds and this interrupts reactions occurring on the surface of AuNPs [37]. Therefore, the catalytic activity was decreased when organic molecules bound on the surface of AuNPs.

## 4. Conclusions

A new online CL sensing method was proposed for online synthesis of AuNPs with a 3D hydrodynamic focusing microreactor and direct characterization of the catalytic activity in the flow. AuNPs were synthesized through a single-phase reaction using glucose as reducing agent and stabilizer at room temperature. The property of AuNPs was easily controlled by tuning concentration of reagents in a 3D microreactor during synthesis. The catalyst property of synthesized AuNPs was characterized by microfluidic CL sensing of luminol and NaOCl. With optimized parameters of synthesis, the CL signal was enhanced 171 times. Without adding another stabilizer, AuNPs were stable for more than one month. Two kinds of reagents were used to change the property of AuNPs and to investigate their effect on catalytic activity. The addition of salt could cause aggregation of synthesized AuNPs and the CL signal for the luminol–NaOCl–AuNPs sensing system was greatly enhanced because of the change of surface charge property. Glutathione was applied as an example of a molecule which binds on the surface of AuNPs. The catalytic activity of AuNPs was decreased and the extent of inhibition was related to the concentration of glutathione. This method offers a good way to confirm optimal synthesis condition of AuNPs for a certain CL sensing application. Researchers can use their own synthesis methods and CL systems according to specific applications. The synthesized AuNPs with good catalyst property can be applied in flow-based CL microarrays instead of enzymes [43,44]. Another potential application is labeling AuNPs with luminol for ultrasensitive CL-based chemical analyses [45]. The quenching effect can also be utilized to detect molecules with special functional groups [46]. For specific detection, AuNPs can bind with antibodies or aptamers [47,48].

## Data Availability

Not applicable.

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
