# Peer review of "Integration of 3D Hydrodynamic Focused Microreactor with Microfluidic Chemiluminescence Sensing for Online Synthesis and Catalytical Characterization of Gold Nanoparticles"

_sensors, 2021, doi:10.3390/s21072290_

Round 1

Reviewer 1 Report

The paper can be published after minor revision reflecting comments inserted as yellow notes into pdf of submitted manuscript

Author Response

  1. The paper can be published after minor revision reflecting comments inserted as yellow notes into pdf of submitted manuscript.

Thanks for your patience and kindness. We appreciate your nice correction of typos and inappropriate expressions. We have accepted all your suggestions and have revised it based on your helpful comments.

Reviewer 2 Report

The work is devoted to integration of 3D hydrodynamic focused microreactor with microfluidic chemiluminescence sensing for online synthesis and catalytical characterization of gold nanoparticles. Authors synthesized AuNPs through a single-phase reaction using glucose as reducing agent and stabilizer in room temperature. The catalyst property of synthesized AuNPs were characterized by microfluidic CL sensing of luminol and NaOCl. The work is of interest because the synthesized AuNPs can be applied in flow-based CL microarrays instead of enzyme. The  labeling AuNPs with luminol can be used for ultrasensitive CL-based chemical analyses. The article looks like a short communication and may be published after major revision.

Notes:

  1. Authors should avoid any abbreviations in the Abstract of the article.
  2. Practical applications of gold nanoparticles should be added in the introduction to the article. From what point of view, it is interesting to obtain these particular nanoparticles? This should be reflected in the introduction by authors.
  3. Authors synthesized AuNPs depending on reagent concentration. Why did not author apply the common methods for characterization of these nanoparticles, for example, Dynamic light scattering or electron microscopy? What size AuNPs are formed? Does the ratio of reagents influence on the size of the formed nanoparticles?
  4. Moreover, the stability of synthesized AuNPs should be also confirmed by Dynamic light scattering method.

Author Response

  1. Authors should avoid any abbreviations in the Abstract of the article.

Thank you for this remark. We have changed all abbreviations in the Abstract.

  1. Practical applications of gold nanoparticles should be added in the introduction to the article. From what point of view, it is interesting to obtain these particular nanoparticles? This should be reflected in the introduction by authors.

Thank you for bringing up this point. We have added practical applications of AuNPs in the introduction (line64-74) as shown below.

‘Although the most effective AuNPs for a specific CL reaction have not been determined, AuNPs have been successfully used in many practical applications, as shown in Table 1. Even though 38 nm AuNPs have been proved to show better catalytic activity in the luminol - H2O2 CL system[1], they were not used in any of these applications. According to our knowledge, researchers only buy commercial AuNPs or produce them batch-wise. On the one hand, no one can determine which AuNPs exhibit the best catalytic activity. On the other hand, there is no simple way to control the size of AuNPs during the synthesis process. Therefore, it is crucial to develop an online monitoring system which can easily control the property of AuNPs during synthesis and immediately inspect the catalytic CL activity.’

Table 1. Practical applications of AuNPs in different CL systems.

CL system

Sizes of AuNPs

Application

Ref

Luminol - H2O2

13 nm

Detection of L-cysteine in pharmaceutical samples

[2]

Luminol–H2O2

12 nm

Detection of fibrillar fibrin in plasma samples

[3]

Luminol - H2O2

31 ± 3 nm

27 ± 2 nm

Detection of C-reactive protein in serum samples.

[4]

Luminol–H2O2

13 nm

Detection of single-strand DNA in plasma samples

[5]

Lucigenin - H2O2

13 ± 3.0 nm

Detection of histone in serum samples

[6]

Luminol - IO4

4 nm

Determination of polyphenols in tap water.

[7]

Luminol - hydrazine

15 nm

Determination of hydrazine in boiler feed water samples.

[8]

Luminol - AgNO3

13 nm

Detection of antigen with antibody functionalized AuNPs in serum samples

[9]

  1. Authors synthesized AuNPs depending on reagent concentration. Why did not author apply the common methods for characterization of these nanoparticles, for example, Dynamic light scattering or electron microscopy? What size AuNPs are formed? Does the ratio of reagents influence on the size of the formed nanoparticles?

Thank you for these questions. As you mentioned, Dynamic light scattering and electron microscopy are common methods to characterize nanoparticles for size and shape. However, there is no positive or negative correlation between CL signals and sizes of AuNPs. Therefore, we cannot confirm a size that we need before it was characterized in CL chip for the catalytic activity. In this way, the only thing we want to guarantee is that AuNPs can be generated with the synthesis method. And UV-Vis spectroscopy is an easy and economical way to confirm it.[10] After we get the optimal AuNPs for a certain CL reaction, it is not indispensable to confirm the sizes because they can be synthesized under the same conditions all the time. From the relative position of the localized surface plasmon resonance (LSPR), it can be assumed that AuNPs were quasi-spherical with size between 10 and 30 nm.[11] If the synthesized AuNPs are used for further practical applications, more characterization will be applied.

The ratio of reagents will influence the size of formed nanoparticles. In our system, there are three methods to change ratio of reagents. (1) keep concentration of reagents constant and change flow rate of reagents, (2) keep flow rate constant and change concentration of reagents, (3) both concentration and flow rate are changed. For the first method, too high or low flow rate will affect the function of the 3D microreactor. And it is not easy to confirm the constant concentration. For the third method, there are countless situations, and it is not possible to try all of them. Then we used the second method with a proper flow rate and tried a range of concentrations of all reagents. When characterized with CL, the signal changed depending on different concentration of reagents. The size or morphology could have changed.

  1. Moreover, the stability of synthesized AuNPs should be also confirmed by Dynamic light scattering method.

Thank you for this remark. In unstable colloidal dispersions, the plasmon resonance peak shifts to longer wavelengths and broadens as the diameter increases due to aggregation. Therefore, aggregated AuNPs would show red-shifted LSPR peak compared with the peak of fully dispersed freshly synthesized AuNPs. That is the reason why we use UV–Vis spectroscopy and it has been used by many researchers to inspect the stability of AuNPs.[12-16]  The aggregation state of AuNPs can also be revealed by using alternative techniques such as DLS to determine the effective size of AuNPs in solution. However, it is not necessary to use more techniques for the same characterization.  

1            Z. F. Zhang, H. Cui, C. Z. Lai and L. J. Liu. Gold nanoparticle-catalyzed luminol chemiluminescence and its analytical applications. Anal Chem, 2005, 77, 3324-3329.

2            W. Liu, J. Luo, Y. Guo, J. Kou, B. Li and Z. Zhang. Nanoparticle coated paper-based chemiluminescence device for the determination of L-cysteine. Talanta, 2014, 120, 336-341.

3            Y. Zhang, J. Liu, T. Liu, H. Li, Q. Xue, R. Li, L. Wang, Q. Yue and S. Wang. Label-free, sensitivity detection of fibrillar fibrin using gold nanoparticle-based chemiluminescence system. Biosens Bioelectron, 2016, 77, 111-115.

4            M. S. Islam and S. H. Kang. Chemiluminescence detection of label-free C-reactive protein based on catalytic activity of gold nanoparticles. Talanta, 2011, 84, 752-758.

5            Y. Qi, B. Li and Z. Zhang. Label-free and homogeneous DNA hybridization detection using gold nanoparticles-based chemiluminescence system. Biosens Bioelectron, 2009, 24, 3581-3586.

6            Y. He and H. Cui. Label free and homogeneous histone sensing based on chemiluminescence resonance energy transfer between lucigenin and gold nanoparticles. Biosens Bioelectron, 2013, 47, 313-317.

7            S. Li, X. Li, J. Xu and X. Wei. Flow-injection chemiluminescence determination of polyphenols using luminol-NaIO4-gold nanoparticles system. Talanta, 2008, 75, 32-37.

8            A. Safavi, G. Absalan and F. Bamdad. Effect of gold nanoparticle as a novel nanocatalyst on luminol-hydrazine chemiluminescence system and its analytical application. Anal Chim Acta, 2008, 610, 243-248.

9            J. Luo, X. Cui, W. Liu and B. Li. Highly sensitive homogenous chemiluminescence immunoassay using gold nanoparticles as label. Spectrochim Acta A Mol Biomol Spectrosc, 2014, 131, 243-248.

10         P. P. Wadekar. A Review on Gold Nanoprticles Synthesis and Characterization. Universal Journal of Pharmaceutical Research, 2017, 2, 65-69.

11         W. Haiss, N. T. Thanh, J. Aveyard and D. G. Fernig. Determination of size and concentration of gold nanoparticles from UV-vis spectra. Anal Chem, 2007, 79, 4215-4221.

12         L. P. Bressan, J. Robles-Najar, C. B. Adamo, R. F. Quero, B. M. C. Costa, D. P. de Jesus and J. A. F. da Silva. 3D-printed microfluidic device for the synthesis of silver and gold nanoparticles. Microchemical Journal, 2019, 146, 1083-1089.

13         S. Annur, S. J. Santosa and N. Hidayat Aprilita. pH Dependence of Size Control in Gold Nanoparticles Synthesized at Room Temperature. Oriental Journal of Chemistry, 2018, 34, 2305-2312.

14         M. V. Bandulasena, G. T. Vladisavljević, O. G. Odunmbaku and B. Benyahia. Continuous synthesis of PVP stabilized biocompatible gold nanoparticles with a controlled size using a 3D glass capillary microfluidic device. Chemical Engineering Science, 2017, 171, 233-243.

15         V. K. T. Ngo, D. G. Nguyen, T. P. Huynh and Q. V. Lam. A low cost technique for synthesis of gold nanoparticles using microwave heating and its application in signal amplification for detectingEscherichiaColiO157:H7 bacteria. Advances in Natural Sciences: Nanoscience and Nanotechnology, 2016, 7.

16         S. Gomez-de Pedro, M. Puyol and J. Alonso-Chamarro. Continuous flow synthesis of nanoparticles using ceramic microfluidic devices. Nanotechnology, 2010, 21, 415603.

17         C. Duan, H. Cui, Z. Zhang, B. Liu, J. Guo and W. Wang. Size-Dependent Inhibition and Enhancement by Gold Nanoparticles of Luminol−Ferricyanide Chemiluminescence. The Journal of Physical Chemistry C, 2007, 111, 4561-4566.

18         L. Gopfert, M. Elsner and M. Seidel. Isothermal haRPA detection of blaCTX-M in bacterial isolates from water samples and comparison with qPCR. Anal Methods, 2021, 13, 552-557.

19         V. K. Meyer, C. V. Chatelle, W. Weber, R. Niessner and M. Seidel. Flow-based regenerable chemiluminescence receptor assay for the detection of tetracyclines. Anal Bioanal Chem, 2020, 412, 3467-3476.

20         Y. Huang, L. Gao and H. Cui. Assembly of Multifunctionalized Gold Nanoparticles with Chemiluminescent, Catalytic, and Immune Activity for Label-Free Immunoassays. ACS Appl Mater Interfaces, 2018, 10, 17040-17046.

21         L.-Y. Yao, X.-Q. Yu, Y.-J. Zhao and A.-P. Fan. An aptamer-based chemiluminescence method for ultrasensitive detection of platelet-derived growth factor by cascade amplification combining rolling circle amplification with hydroxylamine-enlarged gold nanoparticles. Anal Methods-Uk, 2015, 7, 8786-8792.

22         J. Liu, X. Liu, W. R. Baeyens, J. R. Delanghe and J. Ouyang. A novel probe Au(III) for chemiluminescent image detection of protein blots on nitrocellulose membranes. J Proteome Res, 2008, 7, 1884-1890.

Reviewer 3 Report

The manuscript by Yanwei Wang and Michael Seidel discusses the chemiluminescence of gold nanoparticles in a luminol-NaOCl mixture. The synthesis was performed using glucose/NaOH in a microfluidic approach. The influence of the concentration of NaOH and glucose on the formation of AuNPs was investigated by UV/vis spectroscopy. The spectra indicate the formation of gold colloids, however, without electron microscopy it is a matter of guessing to describe their size and shape. From the relative position of the LSPR one might assume a size between 10 and 30 nm and a quasi-spherical shape, but again, this is purely speculative. A major weak point of this contribution is its missing significance and novelty. As an example, the authors found that the AuNPs suffer from aggregation in the presence of NaCl, however, this has been now known for decades (Fig 8b). Also, the finding that AuNPs show chemiluminescence, while their reagents (water, NaOH, glucose, HAuCl4) do not, is not particularly novel (Fig. 6). It is well known that thiols may strongly bind to Au surfaces. As such it is not at all surprising that glutathione may passivate the surface of AuNPs and hinder the generation of chemiluminescence (Fig. 9). I am convinced that this work has a very poor standing without appropriate electron microscopy data and major revisions of the scientific discussion.

Overall, I cannot recommend a publication of this contribution in the Journal Sensors, owing in part to its lack of novelty (design of experiment, etc.) and in part by its poor presentation quality (bad English, scientific discussion, and figures).

Author Response

Reviewer 3

Comments to the Author

The manuscript by Yanwei Wang and Michael Seidel discusses the chemiluminescence of gold nanoparticles in a luminol-NaOCl mixture. The synthesis was performed using glucose/NaOH in a microfluidic approach. The influence of the concentration of NaOH and glucose on the formation of AuNPs was investigated by UV/vis spectroscopy. The spectra indicate the formation of gold colloids, however, without electron microscopy it is a matter of guessing to describe their size and shape. From the relative position of the LSPR one might assume a size between 10 and 30 nm and a quasi-spherical shape, but again, this is purely speculative. A major weak point of this contribution is its missing significance and novelty. As an example, the authors found that the AuNPs suffer from aggregation in the presence of NaCl, however, this has been now known for decades (Fig 8b). Also, the finding that AuNPs show chemiluminescence, while their reagents (water, NaOH, glucose, HAuCl4) do not, is not particularly novel (Fig. 6). It is well known that thiols may strongly bind to Au surfaces. As such it is not at all surprising that glutathione may passivate the surface of AuNPs and hinder the generation of chemiluminescence (Fig. 9). I am convinced that this work has a very poor standing without appropriate electron microscopy data and major revisions of the scientific discussion.

Thanks for your comment. We would like to answer your questions and remarks separately as shown below.

  1. The spectra indicate the formation of gold colloids, however, without electron microscopy it is a matter of guessing to describe their size and shape. From the relative position of the LSPR one might assume a size between 10 and 30 nm and a quasi-spherical shape, but again, this is purely speculative.

In the CL reaction, there is no positive or negative correlation between CL signals and sizes of AuNPs. In other words, after knowing their precise size and shape, we still cannot determine their catalytic activity. Therefore, the size and shape are not indispensable information in the CL reaction. In our method, we applied online synthesis, and the catalytic performance of the synthesized AuNPs can be changed by different concentrations of reagents. And the catalytic activity of synthesized AuNPs can be directly characterized by CL sensor. After obtaining the optimal AuNPs with perfect catalytic property for a certain CL reaction, the synthesis conditions have already been determined and they can be synthesized repeatably. In any case, the formation of AuNP should be guaranteed for the synthesis method, which is the function of the spectra (as you mentioned). If the synthesized AuNPs are applied in practical applications, more information of shape and size will be supplied by electron microscopy or dynamic light scattering.

  1. A major weak point of this contribution is its missing significance and novelty.

AuNPs have been proved to be enzyme-free catalysts in different CL systems and they have been applied in practical applications. Although some researchers have attempted to find the optimal AuNPs for CL reaction,  the variety of AuNPs being tried is limited.[1] There is no positive or negative correlation between CL signals and sizes of AuNPs. Moreover, for different CL systems, the optimal AuNPs were not the same. [17] In this case, problems arise when researchers want to use AuNPs in a CL system, for example, they do not know what kind of AuNPs (or size) they should buy or synthesize.

We solved this problem by integrating a 3D microreactor with microfluidic CL sensing for online synthesis and the catalytic characterization of AuNPs. Our purpose is to determine the synthesis conditions for producing AuNPs with perfect catalytic activity in a certain CL reaction. This method is easy and can be applied to all online synthesis methods and different CL systems. As far as we know, no one has reported a solution to the same problem. For further applications, the synthesized AuNPs can be applied in flow-based CL microarrays instead of enzyme.[18, 19] In addition, when AuNPs bind to antibodies or aptamers, they can be used for online specific detection. [20, 21]

  1. As an example, the authors found that the AuNPs suffer from aggregation in the presence of NaCl, however, this has been now known for decades (Fig 8b). It is well known that thiols may strongly bind to Au surfaces. As such it is not at all surprising that glutathione may passivate the surface of AuNPs and hinder the generation of chemiluminescence (Fig. 9).

Our purpose is to optimize the synthesis conditions of commonly used AuNPs for better catalytic activity, and the method can be applied for all online synthesis methods. The synthesized AuNPs would have the same properties which have been discovered, such as being aggregated by salt and passivated by glutathione. This is the reason why the synthesized AuNPs have the potential for further applications. Moreover, with better catalytic activity, AuNPs CL system may have higher sensitivity. Further, we applied the salt and glutathione to change the property of AuNPs online and showed the potential for online detections.

  1. Also, the finding that AuNPs show chemiluminescence, while their reagents (water, NaOH, glucose, HAuCl4) do not, is not particularly novel (Fig. 6).

Chloroauric acid-enhanced CL has been reported[22], and the catalytic activity of Au3+ could be better than that of some AuNPs[1].The effect of Au3+ should be considered, as the synthesis was online and concentrations of reagents were changed when we tried to get the best synthesis conditions. Also, it would be better to exclude effect from other synthesis reagents.

References

1            Z. F. Zhang, H. Cui, C. Z. Lai and L. J. Liu. Gold nanoparticle-catalyzed luminol chemiluminescence and its analytical applications. Anal Chem, 2005, 77, 3324-3329.

2            W. Liu, J. Luo, Y. Guo, J. Kou, B. Li and Z. Zhang. Nanoparticle coated paper-based chemiluminescence device for the determination of L-cysteine. Talanta, 2014, 120, 336-341.

3            Y. Zhang, J. Liu, T. Liu, H. Li, Q. Xue, R. Li, L. Wang, Q. Yue and S. Wang. Label-free, sensitivity detection of fibrillar fibrin using gold nanoparticle-based chemiluminescence system. Biosens Bioelectron, 2016, 77, 111-115.

4            M. S. Islam and S. H. Kang. Chemiluminescence detection of label-free C-reactive protein based on catalytic activity of gold nanoparticles. Talanta, 2011, 84, 752-758.

5            Y. Qi, B. Li and Z. Zhang. Label-free and homogeneous DNA hybridization detection using gold nanoparticles-based chemiluminescence system. Biosens Bioelectron, 2009, 24, 3581-3586.

6            Y. He and H. Cui. Label free and homogeneous histone sensing based on chemiluminescence resonance energy transfer between lucigenin and gold nanoparticles. Biosens Bioelectron, 2013, 47, 313-317.

7            S. Li, X. Li, J. Xu and X. Wei. Flow-injection chemiluminescence determination of polyphenols using luminol-NaIO4-gold nanoparticles system. Talanta, 2008, 75, 32-37.

8            A. Safavi, G. Absalan and F. Bamdad. Effect of gold nanoparticle as a novel nanocatalyst on luminol-hydrazine chemiluminescence system and its analytical application. Anal Chim Acta, 2008, 610, 243-248.

9            J. Luo, X. Cui, W. Liu and B. Li. Highly sensitive homogenous chemiluminescence immunoassay using gold nanoparticles as label. Spectrochim Acta A Mol Biomol Spectrosc, 2014, 131, 243-248.

10         P. P. Wadekar. A Review on Gold Nanoprticles Synthesis and Characterization. Universal Journal of Pharmaceutical Research, 2017, 2, 65-69.

11         W. Haiss, N. T. Thanh, J. Aveyard and D. G. Fernig. Determination of size and concentration of gold nanoparticles from UV-vis spectra. Anal Chem, 2007, 79, 4215-4221.

12         L. P. Bressan, J. Robles-Najar, C. B. Adamo, R. F. Quero, B. M. C. Costa, D. P. de Jesus and J. A. F. da Silva. 3D-printed microfluidic device for the synthesis of silver and gold nanoparticles. Microchemical Journal, 2019, 146, 1083-1089.

13         S. Annur, S. J. Santosa and N. Hidayat Aprilita. pH Dependence of Size Control in Gold Nanoparticles Synthesized at Room Temperature. Oriental Journal of Chemistry, 2018, 34, 2305-2312.

14         M. V. Bandulasena, G. T. Vladisavljević, O. G. Odunmbaku and B. Benyahia. Continuous synthesis of PVP stabilized biocompatible gold nanoparticles with a controlled size using a 3D glass capillary microfluidic device. Chemical Engineering Science, 2017, 171, 233-243.

15         V. K. T. Ngo, D. G. Nguyen, T. P. Huynh and Q. V. Lam. A low cost technique for synthesis of gold nanoparticles using microwave heating and its application in signal amplification for detectingEscherichiaColiO157:H7 bacteria. Advances in Natural Sciences: Nanoscience and Nanotechnology, 2016, 7.

16         S. Gomez-de Pedro, M. Puyol and J. Alonso-Chamarro. Continuous flow synthesis of nanoparticles using ceramic microfluidic devices. Nanotechnology, 2010, 21, 415603.

17         C. Duan, H. Cui, Z. Zhang, B. Liu, J. Guo and W. Wang. Size-Dependent Inhibition and Enhancement by Gold Nanoparticles of Luminol−Ferricyanide Chemiluminescence. The Journal of Physical Chemistry C, 2007, 111, 4561-4566.

18         L. Gopfert, M. Elsner and M. Seidel. Isothermal haRPA detection of blaCTX-M in bacterial isolates from water samples and comparison with qPCR. Anal Methods, 2021, 13, 552-557.

19         V. K. Meyer, C. V. Chatelle, W. Weber, R. Niessner and M. Seidel. Flow-based regenerable chemiluminescence receptor assay for the detection of tetracyclines. Anal Bioanal Chem, 2020, 412, 3467-3476.

20         Y. Huang, L. Gao and H. Cui. Assembly of Multifunctionalized Gold Nanoparticles with Chemiluminescent, Catalytic, and Immune Activity for Label-Free Immunoassays. ACS Appl Mater Interfaces, 2018, 10, 17040-17046.

21         L.-Y. Yao, X.-Q. Yu, Y.-J. Zhao and A.-P. Fan. An aptamer-based chemiluminescence method for ultrasensitive detection of platelet-derived growth factor by cascade amplification combining rolling circle amplification with hydroxylamine-enlarged gold nanoparticles. Anal Methods-Uk, 2015, 7, 8786-8792.

22         J. Liu, X. Liu, W. R. Baeyens, J. R. Delanghe and J. Ouyang. A novel probe Au(III) for chemiluminescent image detection of protein blots on nitrocellulose membranes. J Proteome Res, 2008, 7, 1884-1890.

Round 2

Reviewer 2 Report

Authors proposed a new online CL sensing method for online synthesis of AuNPs with a 3D hydrodynamic focusing microreactor and direct characterization of the catalytic activity in the flow. Chemiluminescence assays showed great advantages compared with other optical techniques. Gold nanoparticles have drawn big attention in chemiluminescence analysis systems as enzyme-free catalyst. It was shown that microreactor for gold nanoparticles synthesis and direct coupling with microfluidic chemiluminescence sensing offers a promising monitoring method to find the best synthesis condition of gold nanoparticles for catalytic activity.

Authors have corrected all comments in the paper and quite clearly answered to the questions. I hope these corrections improved the paper and the revised version corresponds to high standards of Sensors. After careful consideration, I think that this article may be published in this view.

Author Response

Thank you for your comments. Your comments and suggestions helped us to significantly improve the manuscript and sort out unclarities on the presentation of our findings.

Reviewer 3 Report

The authors Wang and Seidel have provided a revised version of their contribution and have responded to all comments. The revisions are limited to few alterations of the text (wording) and a new Table 1. Table 1 allows for a comparison of CL systems, AuNP sizes, and proposed applications from various reports in literature. In summary, I have to say that the modifications and the answers are not very convincing. In my previous report, I concluded that “this work has a very poor standing without appropriate electron microscopy data and major revisions of the scientific discussion.” The authors have indicated that they do not see the need for providing information on the size and shape of the produced nanoparticles. They explain to their expect neither “a positive or negative correlation between CL signals and sizes of AuNPs”. I am not in the position to challenge their reasoning, however, from my point of view it is bad scientific practice to withhold characterization data from the reader. I am convinced that it would benefit the scientific merit of this paper to provide all necessary information of the studied system, and clearly particle size and shape is crucial for many applications. Thus, I still feel unable to endorse an acceptance for publication.

Author Response

We would like to thank reviewer 3 for his positive evaluation of our work as well as for his constructive critical input. We addressed all comments/questions raised by the reviewer as you can see see in the documentd, made according changes in the revised manuscript, and we have added a REM image of the gold nanoparticles as suggested from reviewer 3. We believe that the revised version is better understandable now and can be of good use for the readership of Sensors.
